# Glazed Photovoltaic-thermal (PVT) Collectors for Domestic Hot Water Preparation in Multifamily Building

**Nikola Pokorny * and Tomáš Matuška**

University Centre for Energy Efficient Buildings, Czech Technical University in Prague, 273 43 Bustehrad, Czech Republic; tomas.matuska@fs.cvut.cz

\* Correspondence: nikola.pokorny@cvut.cz; Tel.: +420-777-048-835

**Abstract:** Photovoltaic–thermal collector generates electrical and thermal energy simultaneously from the same area. In this paper performance analysis of a potentially very promising application of a glazed photovoltaic–thermal collector for domestic hot water preparation in multifamily building is presented. Solar system in multifamily building can be installed on the roof or integrated in the façade of the building. The aim of this simulation study is to show difference of thermal and electrical performance between façade and roof installation of a glazed photovoltaic-thermal collectors at three European locations. Subsequently, this study shows benefit of photovoltaic-thermal collector installation in comparison with side-by-side installation of conventional system. For the purpose of simulation study, mathematical model of glazed photovoltaic-thermal collector has been experimentally validated and implemented into TRNSYS. A solar domestic hot water system with photovoltaic–thermal collectors generates more electrical and thermal energy in comparison with a conventional system across the whole of Europe for a particular installation in a multifamily building. The specific thermal yield of the photovoltaic–thermal system ranges between 352 and 582 kWh/m$^2$. The photovoltaic–thermal system electric yield ranges between 63 and 149 kWh/m$^2$. The increase in electricity production by the photovoltaic–thermal system varies from 19% to 32% in comparison with a conventional side-by-side system. The increase in thermal yield differs between the façade and roof alternatives. Photovoltaic-thermal system installation on the roof has higher thermal yield than conventional system and the increase of thermal yield ranges from 37% to 53%. The increase in thermal yield of façade photovoltaic-thermal system is significantly higher in comparison with a conventional system and ranges from 71% to 81%.

**Keywords:** PVT collector; integrated PVT; energy-active façade; solar energy

## 1. Introduction

Photovoltaic–thermal (PVT) collector represents a technology which combine a solar thermal collector and a photovoltaic module in a single component. Simultaneous generation of power and heat from a limited area of the building envelope (roof, façade) can maximize the fraction of renewable energy source utilization of the energy supply for buildings. The market for PVT collectors is promising; the total installed capacity reached 1 million m$^2$ in 2018, but the share of glazed PVT collectors is very low [1]. At the moment unglazed PVT collectors are widely available on the market. Unglazed PVT collectors are advantageous in terms of electrical performance but heat generation is at low temperature levels. Glazed PVT collectors allow sufficient temperature levels for domestic hot water preparation, because additional glazing reduces thermal losses. Thermal output of glazed PVT collector is comparable with conventional solar thermal collector. However, electrical performance

of glazed PVT collector is reduced in comparison to conventional photovoltaic (PV) panel due to additional glazing and higher operation temperatures. The principal barrier for the glazed PVT collector application is the low resistance of ethylene-vinyl acetate (EVA) lamination of PV cells to excessive thermal exposure. The risk of EVA delamination increases strongly for temperatures over 135 °C [2], while the stagnation temperature for glazed PVT collectors could reach from 120 to 180 °C. At such high temperature, EVA lamination decomposes to acetic acid. This phenomenon causes the delamination, corrosion of PV cell contacts, and degradation of layer transparency. One possibility of how to prevent stagnation is to use different strategies to control heat losses of the collector. In the scientific literature, several approaches exist for how to avoid stagnation by increasing significantly heat losses during the stagnation period, focusing on solar thermal collectors [3] and glazed PVT collectors [4]. The concept of the glazed PVT collector presented in this paper uses polysiloxane gel as a thermally resistant encapsulant for PV cells [5]. The main benefit of the gel is a wide range of operation temperatures (from −60 to +250 °C) [6]. The disadvantage of the encapsulation of crystalline PV cells into gel is the nonstandard process of lamination. Currently, manufacturers use solely EVA foil for lamination of crystalline PV cells.

Solar domestic hot water (SDHW) preparation with PVT collectors is deeply analysed topic, especially in the context of single-family houses. A comparison between a glazed and unglazed PVT system for a single-family house was studied in [7]. Four different mathematical models of a PVT collector were evaluated for SDHW applications in a single-family house [8]. One analysis was focused on a comparison of the conventional side-by-side system with a PVT system consisting of unglazed PVT collectors; this system was evaluated for three climatic locations [9]. In the case of multifamily buildings, unglazed PVT collectors can be applied for preheating of domestic hot water (DHW) [10]. However, a larger solar thermal fraction of the SDHW system in a multifamily building can be achieved with glazed PVT collectors [5]. Another option for PVT collectors is the application in a combi system not only for preparation of domestic hot water (DHW) but also for heating. The combi system can be realized by different types of PVT systems in combination with a ground-source heat pump [11]. A measured combi system with unglazed PVT collectors and a ground-source heat pump was evaluated in terms of utilization of the coefficient of performance of the heat pump [12]. In the context of a combi system, one exergetic analysis was focused on building integrated unglazed PVT collectors in combination with air source heat pump [13]. The comparison of performance between different PVT systems with glazed and unglazed PVT collectors was evaluated [14]. No study has been found in the literature with regards to a glazed PVT collector integrated into a façade in comparison with a usual roof installation.

This paper provides a performance comparison between façade integration and roof installation of glazed PVT collectors for DHW preparation in a multifamily building. A specific condition for the glazed PVT collector application is the integration into a façade envelope as a prefabricated unit. The PVT collector is a part of the façade construction, and the concept of the envelope, considered as an energy loss in the past, turns into an energy-active envelope. In the paper, the developed and fabricated prototypes of glazed PVT collector in standard frame and integrated alternative are presented; see Figure 1. The analysis is focusing on the multifamily building because of high heat demand and limited area on the roof. In this application, the glazed PVT collector could be competitive not only from an economical point of view but also an ecological one. Performance analysis for a multifamily building with developed detailed mathematical model of the glazed PVT collector has been carried out for three different European climates and compared with a conventional solar energy system consisting of photovoltaic panels and solar thermal collectors.

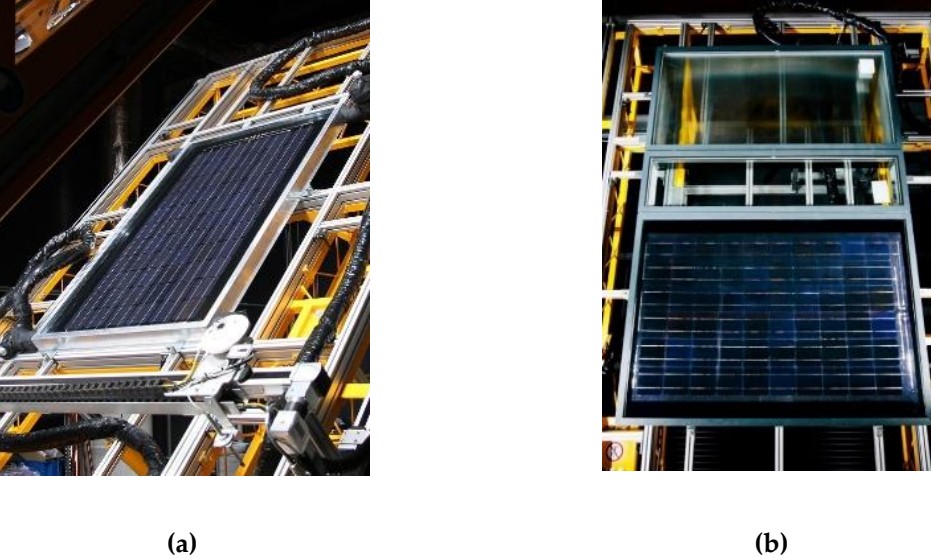

| (a) | (b) |

**Figure 1.** Separate PVT collector in a standard frame (**a**) and façade-integrated PVT collector (**b**) on the test stand.

## 2. Concept of Glazed PVT Collector

The glazed PVT collector concept, both in a separate and façade-integration alternative, is based on a sandwich unit consist of double glazing and copper absorber plate. Monocrystalline PV cells are encapsulated in the polysiloxane gel layer which is situated between the double glazing and copper sheet with a pipe register; see Figure 2. Prototypes of glazed PVT collectors in both versions were fabricated. The gap between glass panes is filled with argon. The absorber was manufactured from a copper sheet soldered to the copper pipe register. The absorber was insulated on the back and lateral sides. The sandwich unit can be integrated into a standard collector frame made of aluminium profiles or integrated into a curtain walling façade element. Detailed geometrical, optical and thermal properties of both prototypes are in Table 1.

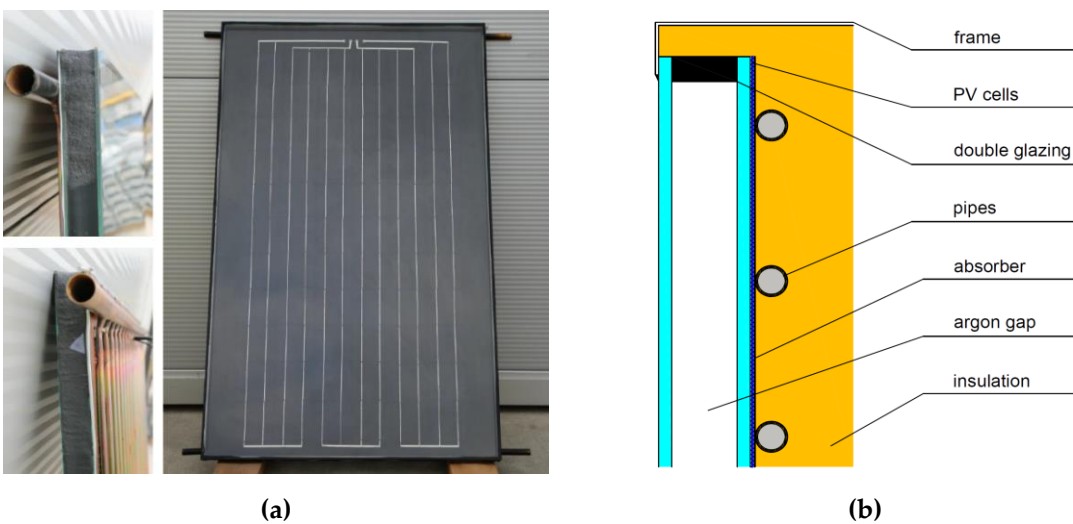

| (a) | (b) |

**Figure 2.** Fabricated PVT sandwich unit without frame insulation (**a**) and layout of the prototype (**b**).

**Table 1.** Tested prototypes of PVT collectors.

| Parameters | Separate PVT | Integrated PVT | Unit |
|---|---|---|---|
| **Geometrical Properties** | | | |
| Gross area | 1.65 | 1.56 | m$^2$ |
| Aperture area | 1.55 | 1.37 | m$^2$ |
| Gap between glazing and front side of PVT absorber | 24 | 24 | mm |
| Gap between back side of PVT absorber and frame | 5 | 5 | mm |
| Thickness of glass | 4 | 4 | mm |
| Thickness of insulation (back side) | 40 | 160 | mm |
| **Properties of the sheet and tube absorber** | | | |
| Thermal conductivity of the absorber plate | 350 | 350 | W/m.K |
| Thickness of absorber plate | 0.2 | 0.2 | mm |
| Number of riser pipes | 20 | 18 | - |
| Distance between the pipes | 50 | 50 | mm |
| Internal diameter of riser pipes | 7.2 | 7.2 | mm |
| **Optical and thermal properties** | | | |
| Normal solar transmittance of glass cover | 0.92 | 0.91 | - |
| Absorptance of PVT absorber | 0.81 | 0.91 | - |
| Front surface emissivity of PVT absorber | 0.3 | 0.85 | - |
| Thermal conductivity of glass cover | 0.8 | 0.8 | W/m.K |
| Thermal conductivity of insulation (mineral wool) | 0.04 | 0.04 | W/m.K |
| **Properties of PV part** | | | |
| Zero-loss electrical efficiency of PV part without cover glass pane (related to PV area) | 12.9 | 15.3 | % |
| Temperature coefficient of PV cell efficiency | 0.44 | 0.44 | %/K |
| Gas layer between glazing | Argon | Argon | - |
| Packing factor | 0.62 | 0.6 | - |

The size of a separate collector was 1.02 m × 1.62 m, with a gross area of 1.65 m$^2$. In the case of the separate collector, a low-emissivity (low-e) coating was applied to the inner glass surface of the double glazing. By application of a low-e coating, it is possible to considerably decrease radiative heat losses [15]. However, this leads to a reduction in transmittance and therefore to a reduction of the amount of incident radiation on PV cells. In total, 6 × 11 monocrystalline PV cells sized 125 × 125 mm were used. The PV cell reference efficiency declared by the manufacturer was 18.8% under STC (standard test conditions). The gross area of the PVT collector was filled with PV cells by 62% (packing factor related to gross area).

The size of the collector unit integrated into a façade element was 1.50 m × 1.38 m, with a gross area of 1.56 m$^2$. Monocrystalline cells were identical to a separate collector alternative, but only 60 PV cells were used. Packing factor was achieved similar (60%). Integration of the PVT sandwich unit into a façade module resulted in a total insulation thickness behind the absorber of 160 mm. No low-e coating has been applied for glazing in the prototype, which was tested under steady-state conditions and presented in this paper. Nevertheless, a comparison of two integrated PVT collectors with and without a low-e coating was carried out under outdoor climatic conditions and published in [16]. Thanks to the low-e coating, the 10% increase in thermal production was evaluated with a low decrease in electrical production.

Prototypes were tested at the Solar Laboratory in UCEEB CTU (University Centre for Energy Efficient Buildings of Czech Technical University) in Prague with the use of indoor solar simulator. The lamp field consisted of eight metal halide lamps (each 4.6 kW). The indoor facility allowed for achieving the solar irradiance of up to 1200 W/m$^2$, with the homogeneity of ± 10% in the interior environment in the area of 2.0 × 2.4 m; irradiance was close to the solar spectrum (AM 1.5). In Figure 1, two prototypes of integrated and separated glazed PVT collectors on the test stand are shown.

The separated PVT collector was fixed with a slope of 45° and the integrated one with the slope of 90°. The thermal performance was evaluated according to EN ISO 9806 [17]. Artificial wind was applied to the collector plane. Thermal output was evaluated for five different inlet temperatures of water (from 16 to 91 °C). Characteristics of thermal efficiency were tested in two modes. In first mode, the PV part was maintained at a maximum power point (MPP). In second open-circuit mode, the PVT collector thermal performance was tested without electric load. Inlet water temperature to the collector, outlet water temperature from the collector (accuracy ± 0.04 K) and ambient air temperature (accuracy ± 0.2 K) were measured by PT100 sensors. The mass flow rate was measured by a Krohne Optimass 7000/MFC300 (accuracy ± 1%). Incident solar irradiance was measured by the Kipp and Zonen SMP11-A pyranometer (accuracy ± 1.35%). The Metrel MI 3108 unit was used for measurement of electric power (accuracy ± 2.5%).

Figure 3 shows the tested thermal and electrical characteristics for a separate PVT collector module with a low-e coating. On the *y*-axis, there was a reduced temperature difference, where $t_e$ [°C] is ambient air temperature, $t_m$ [°C] is mean fluid temperature of the heat transfer fluid and $G$ [W/m$^2$] is incident solar irradiance on the PVT collector area. Figure 4 shows the thermal and electrical characteristics for the integrated PVT collector without low-e coating.

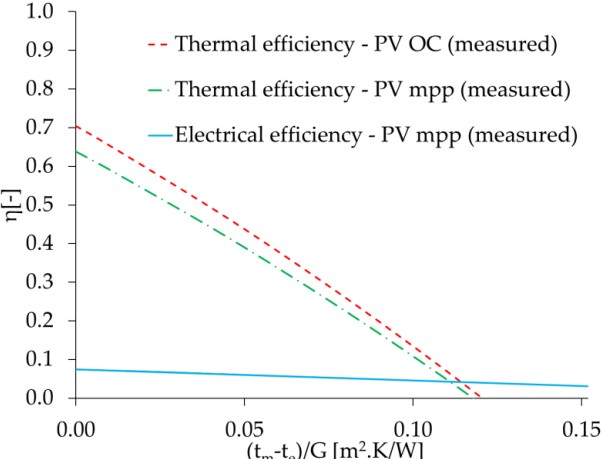

**Figure 3.** Measured thermal and electrical characteristics for the separate PVT collector (related to gross area).

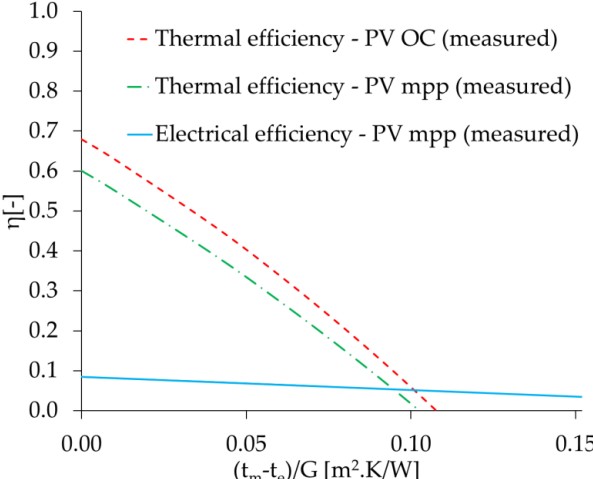

**Figure 4.** Measured thermal and electrical characteristics for the integrated PVT collector (related to gross area).

In the case of the separate alternative, the open-circuit conditions were the following: the global irradiance was maintained at the average value of 918 W/m$^2$ during the test. Ambient temperature was maintained at 18.9 °C. The collector tilt angle was 45°. The PVT collector zero-loss thermal efficiency was evaluated at 70%. The stagnation temperature in open-circuit conditions was determined to be 175 °C. Test conditions with electric load were following: the global irradiance was maintained at the average value of 915 W/m$^2$ during the test. Ambient temperature was maintained at 18.7 °C. The PVT collector tilt angle was 45°. The PVT collector zero-loss thermal efficiency was evaluated at 64% and the zero-loss electrical efficiency at 7.4% (both efficiencies are related to gross area).

Test conditions without electric load (open-circuit) for the integrated collector were the following: the global irradiance was maintained at the average value of 931 W/m$^2$ during the test. The average ambient temperature during the test was maintained at 16.4 °C. The integrated PVT collector tilt angle was 90°. Zero-loss thermal efficiency of the PVT collector was evaluated at 68% (related to gross area). The stagnation temperature in open-circuit conditions was determined to be 161 °C. Test conditions with electric load were following: the global irradiance was maintained at the average value of 924 W/m$^2$ during the test. The average ambient temperature was maintained at 18.2 °C. The collector tilt angle was 90°. The PVT collector zero-loss thermal efficiency was evaluated at 60% and the zero-loss electrical efficiency at 8.4% (both efficiencies are related to gross area).

In Table 2, results of indoor measurement are shown for the mode with MPP tracking. Values of coefficients of thermal efficiency were evaluated in the expected range. However, zero-loss electrical efficiency of the module was expected to be around a value of 8.9%. The main reason for the decrease was attributed to connection losses between PV cells. When the module's electrical efficiency was compared with other research focused on the development of a glazed PVT collector with a low-e coating [18] or research focused on the comparison of a glazed PVT with and without low-e coating [19], the electrical efficiency of the tested PVT collector was lower, but it was mainly influenced by the packing factor of the collector. The original idea was to fabricate a PVT collector with the same gross area as commercially available PV panels. Due to this fact, the packing factor was unfortunately much lower because of standardized fabrication of crystalline PV cells. The packing factor is possible to increase in the future with a change of dimensions of the collector. Thermal efficiency was comparable with other research that was done in the field of glazed PVT development.

**Table 2.** Results of measurement under steady-state conditions in hybrid mode (related to gross area).

| Parameters | Separate PVT | Integrated PVT | Unit |
|---|---|---|---|
| Zero-loss thermal efficiency $\eta_{0,G}$ | 0.639 | 0.601 | - |
| Linear heat loss coefficient $a_{1,G}$ | 4.644 | 4.831 | W/m$^2$.K |
| Quadratic heat loss coefficient $a_{2,G}$ | 0.007 | 0.009 | W/m$^2$.K$^2$ |
| Module zero-loss electrical efficiency $\eta_{e,G}$ | 0.074 | 0.084 | - |
| Stagnation temperature | 171 | 156 | °C |

## 3. Simulation Analysis of SDHW System for Multifamily Building

Combination of high heat demand and limited roof area is convenient for application of glazed PVT collectors. Buildings such as multifamily buildings, hotels have a limited roof area and uniform heat and electricity load during the year. Therefore, SDHW preparation for multifamily buildings can be a target application where glazed PVT collectors could be competitive in comparison with a combination of conventional technologies (photovoltaics, solar thermal collectors).

### 3.1. Mathematical Model of Glazed PVT Collector

Currently, steady-state and dynamic models of glazed PVT collectors are known in the scientific literature. A detailed steady-state analytical model was used for optimization of absorber geometry [20]. Four different mathematical models were compared in terms of thermal yield [8]. An explicit dynamic model was developed for a glazed PVT collector based on the control-volume finite-difference

approach [21]. A 3D dynamic model considering nonuniform temperature distribution on the PV area was developed and validated [22]. An analogous 3D dynamic model was validated in a SDHW system [23]. Nevertheless, none of these models are available in TRNSYS (Transient System Simulation Tool). In TRNSYS software for system simulations, there is a type 50b [24] model for a glazed PVT collector that does not take into account a detailed design of the collector and dependency of important PVT collector parameters on climate and operation conditions (fin efficiency factor, collector heat loss coefficient, etc.). Thermal output from type 50b is not reliable. Due to this fact, a new detailed model for a glazed PVT collector was developed with the use of the Florschuetz approach [25]. The model is based on detailed energy flow balance of a PVT absorber, expanded for photovoltaic conversion. The model has been implemented in TRNSYS as a new type to allow PVT system simulations. The model solves the external and internal energy balance of the PVT absorber. The external balance solves heat transfer from the absorber to the ambient, and the internal balance solves heat transfer from the absorber to the heat transfer fluid. Both balances proceed in the iteration loop. The mathematical model has been developed with steady-state [26] and dynamic modes [27], but for annual simulations, the steady-state is sufficient. However, the mathematical model has some simplifications. The model does not consider the influence of inhomogeneous temperature distribution at the absorber on PV cell efficiency. Nevertheless, the important advantage of the implemented model is the possibility to define a number of design parameters of PVT collector configuration: electrical and thermal properties of PV cells, geometry, thermophysical properties of materials used in the PVT collector, etc. Inputs to the model are following: climate and operation conditions. The main outputs from the model are thermal output and electric power, outlet liquid temperature from the collector, and PVT absorber temperature. Validation of the mathematical model based on the testing allowed to consider a low-emissivity coating for both investigated alternatives in the next chapter: a separate PVT collector on the roof and a PVT collector integrated into the façade module.

Mathematical model validation for the separate PVT collector with a low-e coating is further presented in the paper. To validate the mathematical model, the thermal and electrical efficiency characteristics have been modelled and compared with data from steady-state testing. A previously described test in hybrid mode under steady-state conditions was used for the validation. There was some natural uncertainty in the parameter data for the model, e.g., real thermal conductivity of the insulation, real transmittance of the cover glazing, real absorptance and emissivity of the full absorber area, etc. Most of the used input parameters are in Table 1. For example, uncertainty of the thermal conductivity value for the insulation could be considered about 10% if not determined by special testing, transmittance of the cover glazing was considered with uncertainty about 2%, emissivity of the absorber 2% according to the datasheet of the manufacturer, electrical efficiency of the PV part 5%, etc. Therefore, the thermal and electrical characteristics were modelled as two boundary lines expressing the full range of all parameter uncertainty and create the model uncertainty band for a given PVT collector. This band could be diminished if there is knowledge of the parameters with better precision. The experimental data derived for steady-state laboratory tests of the PVT collector in hybrid mode lay within the model uncertainty band (see Figures 5 and 6, blue lines). Moreover, Figures 5 and 6 also show measurement uncertainty. The measurement uncertainty of thermal efficiency was determined by the methodology described in the standard for testing of solar thermal collectors, ISO 9806 [17]. The experimental results of testing are necessary to determine the three-parameter steady-state model, which describes the thermal collector behaviour and it is widely used in practice for energetic calculation. A detailed description of different aspects of calculation of uncertainties in solar thermal collector testing can be found in [28]. Expanded uncertainty for electrical power was carried out by a standard procedure, with consideration of wattmeter accuracy.

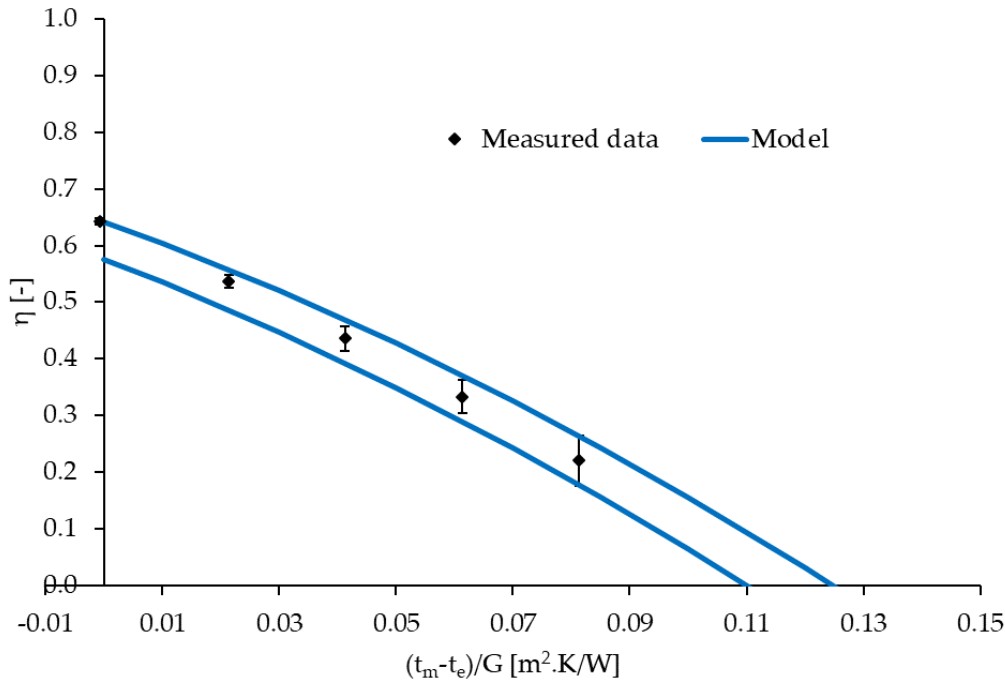

**Figure 5.** Validation of thermal characteristics of glazed PVT collector.

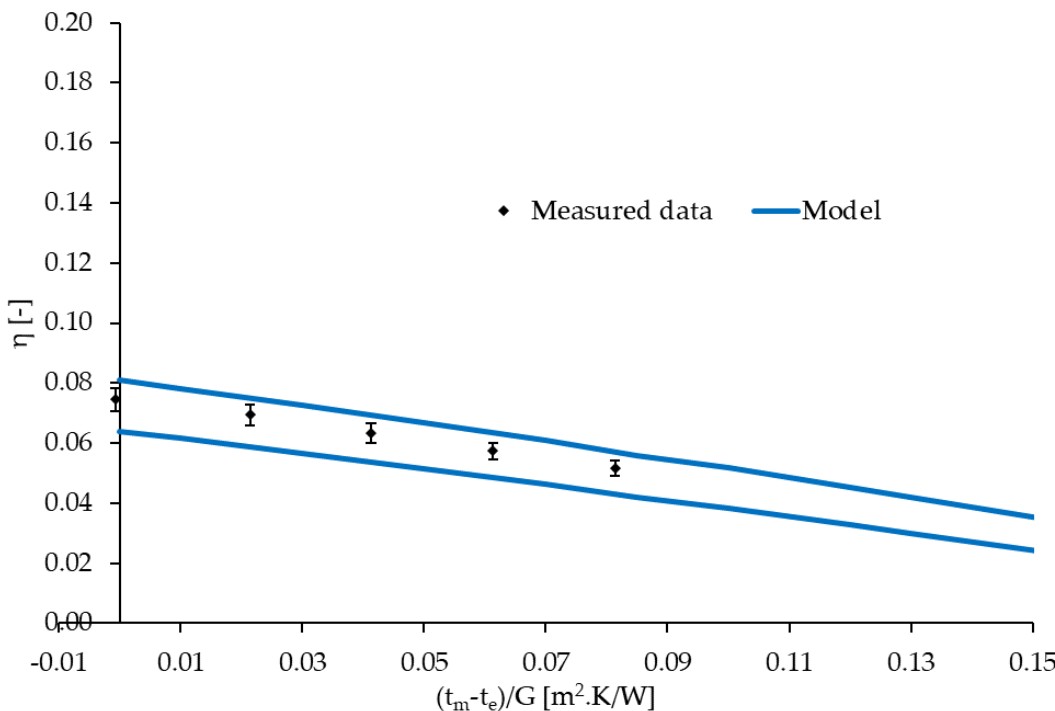

**Figure 6.** Validation of electrical characteristics of glazed PVT collector.

*3.2. SDHW System in Multifamily Building*

An SDHW system for a multifamily building has been considered for performance analysis. The multifamily building has 45 flats and 100 occupants. The total floor area of the building was 475 m$^2$, and the south façade area was 630 m$^2$. Gross floor size of the building was 25 m × 19 m, with a total height of 25.2 m (9 floors). Solar heat was used merely for DHW preparation; solar electricity was primarily used for the building appliance load. DHW demand in the multifamily building was

considered 116 MWh/a. Electricity demand in the building was considered 96 MWh/a. The load profile of DHW preparation and electricity demand for one day is shown in Figures 7 and 8.

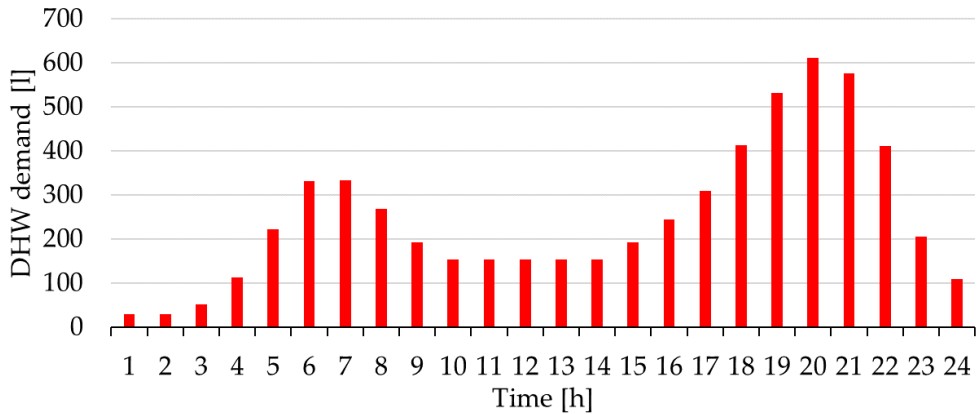

**Figure 7.** Load profile of domestic hot water.

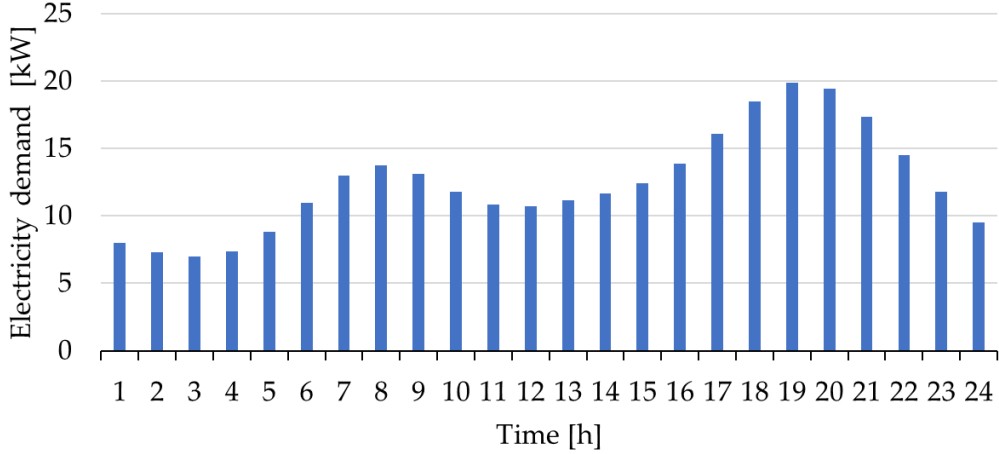

**Figure 8.** Profile of electricity demand for one day.

The analysis has been carried out for three different climate zones in Europe with different annual solar irradiation: Madrid (1864 kWh/m$^2$.a), Prague (1115 kWh/m$^2$.a) and Helsinki (1153 kWh/m$^2$.a). Three different alternatives of solar collector placement have been considered: roof (R), integrated into the façade on all floors (F9) and integrated into the façade on only five upper floors (F5). Solar systems with glazed PVT liquid collectors were compared with a combination of conventional solar thermal collectors and PV panels in the same area (50% of solar thermal collectors and 50% of PV panels). The ratio between solar thermal collectors and PV panels was chosen only for better clarity of the paper. In another study, different ratios of conventional systems could achieve higher energy production [5]. A conventional solar energy system consist of the state-of-the-art solar thermal collectors and PV modules with identical monocrystalline PV cells, as used in PVT collectors.

The scheme of the solar energy system with PVT collectors is shown in Figure 9. Main components of the investigated system were designed according to the area of the solar thermal part; see Table 3. Solar thermal and PVT collectors were considered with a slope of 45° for roof installation and 90° for integration into the façade. For the side-by-side alternative, an inclination of 30° has been used for PV panels (to maximize electricity production). Moreover, the following parameters were considered: orientation to the south, a heat exchanger, insulated pipes of the collector loop and an insulated solar DHW storage tank. Dimensions of the collector loop pipes were based on a specific mass flow rate of 15 kg/h.m$^2$ of the collector area (low-flow solar system). The hydraulic connection between collectors was considered parallel. The length of the solar collector loop in an outdoor environment was 80 m;

the length of pipes inside the building was 80 m. For the façade-integrated alternative, the collector loop pipes were considered fully installed in an indoor environment. The maximum temperature in the storage tank was considered to be 85 °C. The required domestic hot water temperature was considered to be 55 °C. Cold water temperature was considered to be 10 °C. Controller temperature difference for the solar circuit pump was 7 °C (on) and 2 °C (off). Efficiency of the solar plate heat exchanger was considered to be 80%. Solar tank volume was calculated from the specific value of 50 L/m$^2$ of the solar thermal collector area (see Table 3). The PV system was a conventional grid-on system with a DC/AC inverter. Electric losses of the system were considered to be 10%. PV electricity generation was assumed to be primarily consumed for the building appliance load.

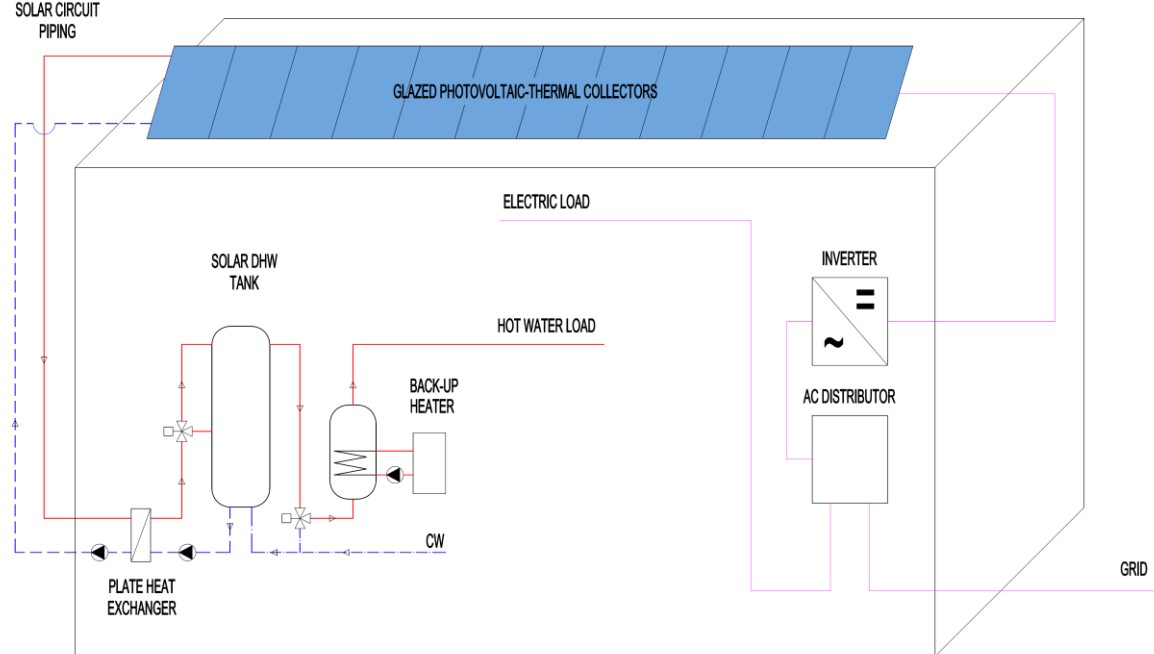

**Figure 9.** Scheme of the solar energy system in a multifamily building.

**Table 3.** Parameters of main components in the solar thermal system.

| Alternative | Solar Thermal Area [m$^2$] | Solar Tank Volume [m$^3$] | Collector Loop Dimension [mm] | Collector Loop Insulation [mm] |
|---|---|---|---|---|
| R—PVT | 165 | 8.3 | 35 × 1.5 | 25 |
| R—50PV50PT | 82.5 | 4.1 | 28 × 1.5 | 25 |
| F9—PVT | 140 | 7 | 35 × 1.5 | 25 |
| F9—50PV50PT | 70 | 3.5 | 28 × 1.5 | 25 |
| F5—PVT | 78 | 3.9 | 28 × 1.5 | 25 |
| F5—50PV50PT | 39 | 1.9 | 22 × 1.5 | 19 |

### 3.3. TRNSYS Simulation Deck

Performance analysis was done in simulation software TRNSYS [24]. The simulation time step was 6 min. Climatic data for a TMY (typical metrological year) from the Meteonorm database (Madrid, Prague, Helsinki) were used. For the SDHW system, the following types available in TRNSYS library were used: solar tank (type 4c), plate heat exchanger (type 91), pipe heat losses (type 709), etc.

The new detailed model for a PVT collector implemented into TRNSYS (type 223) has been used for a simulation of a PVT system. Parameters of both PVT collector alternatives (separated, integrated) are in Appendix A; several parameters were different in comparison to tested variants of the collectors

in Table 1. Parameters were chosen with consideration of the goal and achievable values for future prototypes (e.g., low-e coatings).

In the case of the conventional system, simulation type 832 [29] was used for thermal collector modelling with the following main parameters: $\eta_0 = 0.702$, $a_1 = 3.78$ W/m$^2$K, $a_2 = 0.0135$ W/m$^2$K$^2$ and $c_{eff} = 7000$ J/mK. Electrical power of the conventional PV panel was simulated by type 50b. The packing factor was considered to be 0.9. Reference module electrical efficiency was considered to be 16%.

### 3.4. Installation of Solar Energy System on the Roof

The available area for solar energy system is approximately 35 % of total roof area of the building (475 m$^2$) which is 165 m$^2$; see Figure 10. The conventional installation consisted of solar thermal collectors (82.5 m$^2$) and PV panels (82.5 m$^2$).

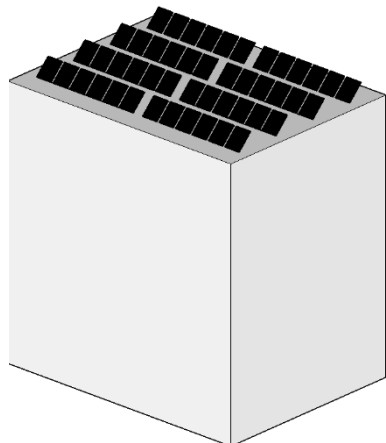

**Figure 10.** Installation of the solar energy system on the roof (R).

### 3.5. Installation of Solar Energy System on the Façade

The roof of the building can have very limited space for solar system installation because the roof is often occupied by other technology systems. Due to this fact, façade integration of the solar energy system was considered as an alternative as well. The first subalternative of façade installation considered the area of 140 m$^2$, which was 22% of the total area of the south wall (fully using the opaque part of the façade for nine floors; see Figure 11a). While the conventional solar energy system installation considered 70 m$^2$ of PV panels and 70 m$^2$ of solar thermal collectors. PVT collectors installation occupies the whole area of 140 m$^2$. In Figure 11, windows are represented by blue colour and solar installations by black colour. To show impact of green vegetation being potentially located in the front of the south façade in the future, only the upper five floors were simulated in the second subalternative (see Figure 11b). The available area of the façade for solar system installation in this case was only 78 m$^2$. Conventional system consists of 39 m$^2$ of PV panels and 39 m$^2$ of solar thermal collectors. PVT collector area was considered 78 m$^2$.

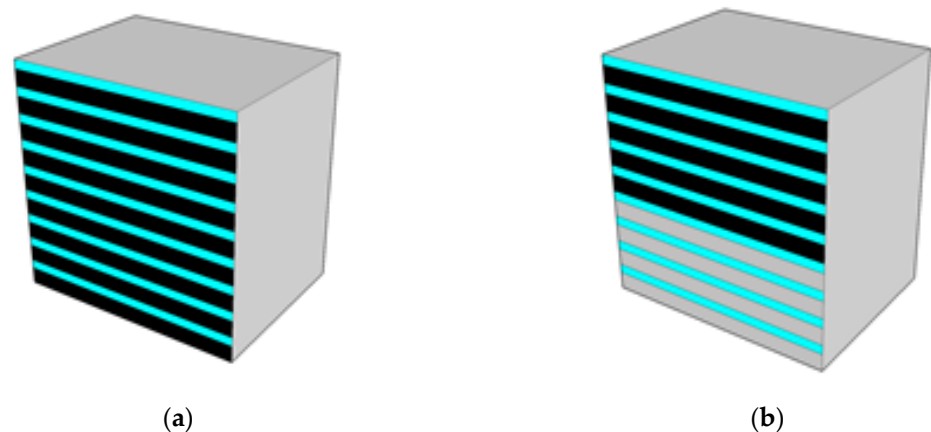

(**a**)                                                          (**b**)

**Figure 11.** Façade solar energy system installed on nine floors (**a**) and on five floors (**b**) of the building.

## 4. Discussion

Simulation study was carried out for three climatic locations. The thermal and electrical yield for every alternative is shown in Figure 12. It is evident that façade installation of PVT collectors achieved significantly lower energy gains compared to roof installation (both heat and electricity).

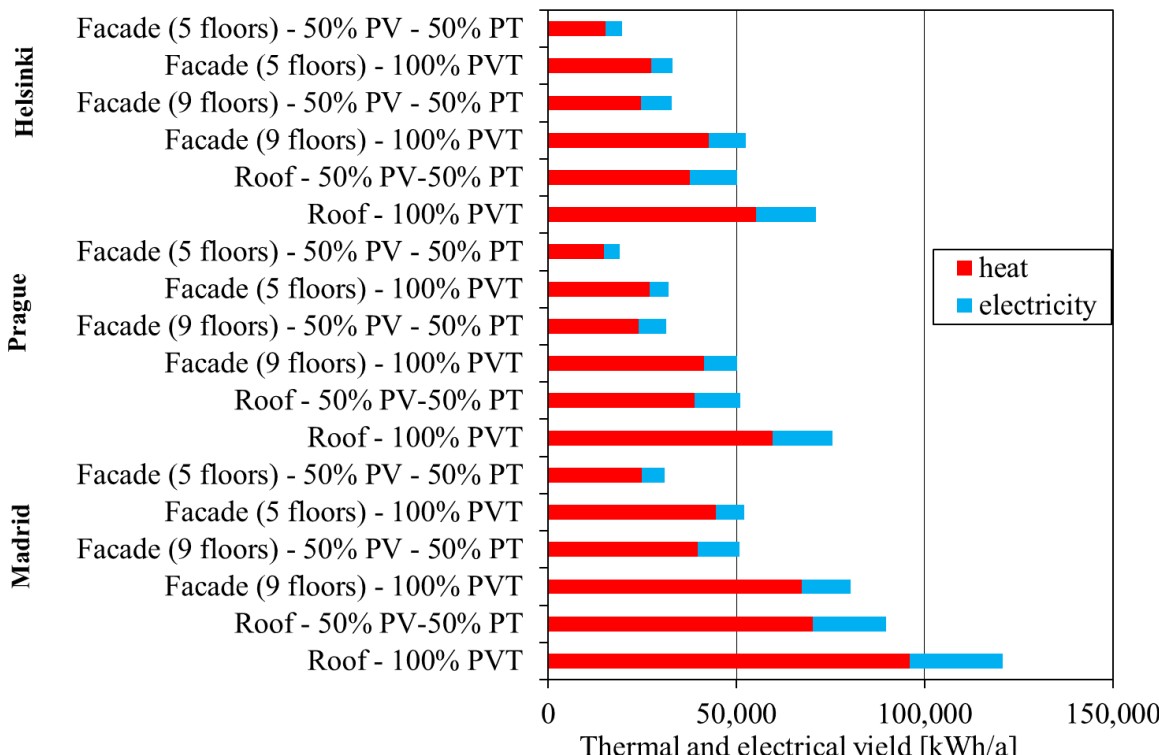

**Figure 12.** Results of the simulation study for a multifamily building.

The PVT systems in every alternative produced more electricity and thermal energy than conventional systems in the three studied locations. The increase in thermal and electrical production thanks to the PVT concept is shown in Table 4. The PVT concept showed much higher thermal production compared to the conventional system, especially in limited façade alternatives, where the increase in thermal production was around 80% (70% in the nine-floor alternative).

**Table 4.** Increase in energy production of PVT alternatives compared to conventional installation.

| Location | Alternative | Heat | Electricity |
|---|---|---|---|
| Madrid | Roof | 37% | 27% |
| | Façade 9 floors | 70% | 19% |
| | Façade 5 floors | 79% | 22% |
| Prague | Roof | 53% | 32% |
| | Façade 9 floors | 72% | 21% |
| | Façade 5 floors | 81% | 23% |
| Helsinki | Roof | 47% | 28% |
| | Façade 9 floors | 72% | 23% |
| | Façade 5 floors | 81% | 25% |

The comparison of the annual yield and solar fraction for particular alternatives and climates is presented in Table 5. Solar thermal and electrical fractions are defined as a ratio between the used energy yield (heat, electricity) and energy demand (heat, electricity) of the building. The specific thermal and electrical yield are related to gross area of the collector. The specific thermal yield of the PVT system ranged between 352 and 582 kWh/m$^2$. The maximum thermal yield and solar fraction were achieved naturally in Madrid. The solar thermal fraction in the case of southern Europe was higher than usually achieved with a solar thermal system. Under normal circumstances, the solar thermal system would be designed with a smaller area of collectors, with consideration of frequent stagnation. The solar fraction for electricity was, in all alternatives, quite low (between 4% and 26%) because of high electricity demand in the building. The maximum solar fraction was achieved again in Madrid. The PVT system's electric yield ranged between 63 and 149 kWh/m$^2$. Monthly usable heat gains from solar systems for the three European locations were compared with monthly DHW demand of the building; see Figures 13–15. It is evident that thermal energy production in the summer months was much higher than the demand of the building, especially for the Madrid location. The highest thermal fraction and solar electricity fraction were achieved with a roof installation of a PVT system. It is obvious that electricity demand in this multifamily building was much higher compared to the maximum production of electricity. To illustrate this, electricity demand in comparison with electricity production of the PVT system on the roof is shown in Figure 16.

**Table 5.** Annual energy yield and solar fraction for different alternatives and climates.

| Alternative | Thermal Yield [kWh/m$^2$] | Electricity Yield [kWh/m$^2$] | Solar Thermal Fraction | Solar Electricity Fraction |
|---|---|---|---|---|
| **Madrid** | | | | |
| Roof—100% PVT | 582 | 149 | 83% | 26% |
| Roof—50% PV—50% PT | 853 | 235 | 61% | 20% |
| Façade (9 floors)—100% PVT | 481 | 94 | 58% | 14% |
| Façade (9 floors)—50% PV—50% PT | 567 | 158 | 34% | 11% |
| Façade (5 floors)—100% PVT | 571 | 96 | 39% | 8% |
| Façade (5 floors)—50% PV—50% PT | 636 | 158 | 21% | 6% |
| **Prague** | | | | |
| Roof—100% PVT | 361 | 96 | 52% | 17% |
| Roof—50% PV—50% PT | 473 | 147 | 34% | 13% |
| Façade (9 floors)—100% PVT | 295 | 63 | 36% | 9% |
| Façade (9 floors)—50% PV—50% PT | 343 | 104 | 21% | 8% |
| Façade (5 floors)—100% PVT | 346 | 64 | 23% | 5% |
| Façade (5 floors)—50% PV—50% PT | 382 | 104 | 13% | 4% |
| **Helsinki** | | | | |
| Roof—100% PVT | 335 | 97 | 48% | 17% |
| Roof—50% PV—50% PT | 456 | 152 | 33% | 13% |
| Façade (9 floors)—100% PVT | 304 | 71 | 37% | 10% |
| Façade (9 floors)—50% PV—50% PT | 354 | 116 | 21% | 8% |
| Façade (5 floors)—100% PVT | 352 | 73 | 24% | 6% |
| Façade (5 floors)—50% PV—50% PT | 390 | 116 | 13% | 5% |

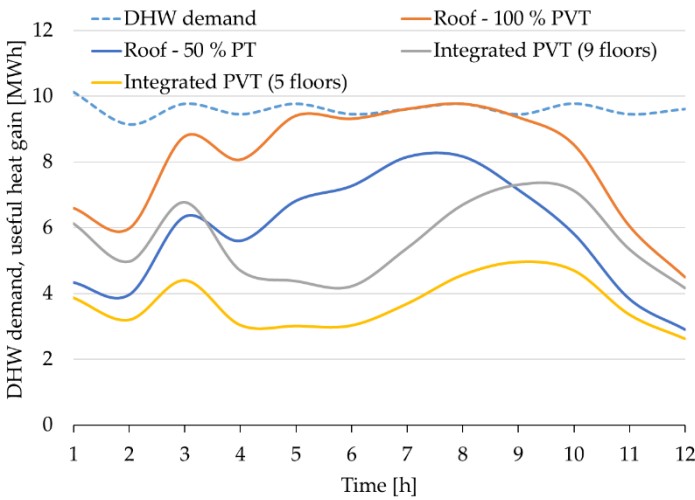

**Figure 13.** Monthly used heat energy and domestic hot water (DHW) demand—Madrid.

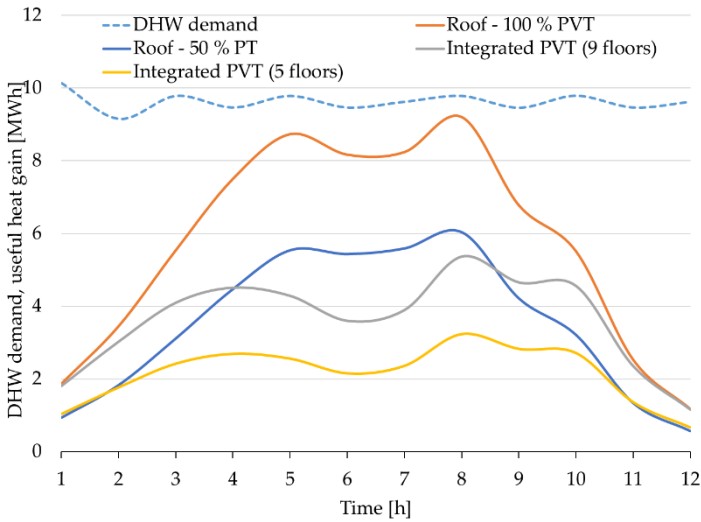

**Figure 14.** Monthly used heat energy and DHW demand—Prague.

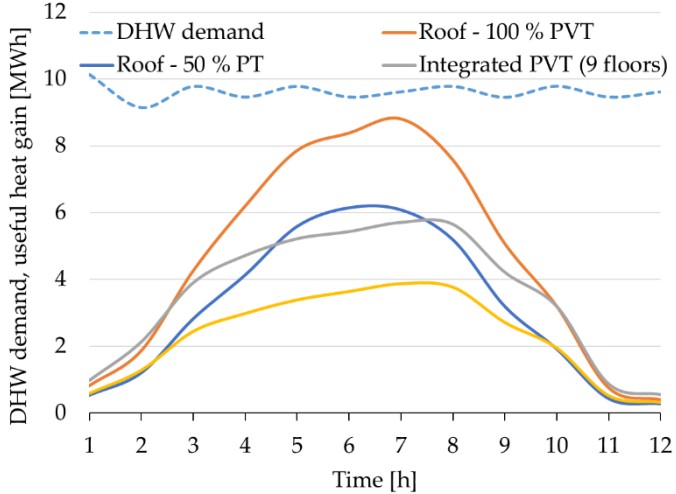

**Figure 15.** Monthly used heat energy and DHW demand—Helsinki.

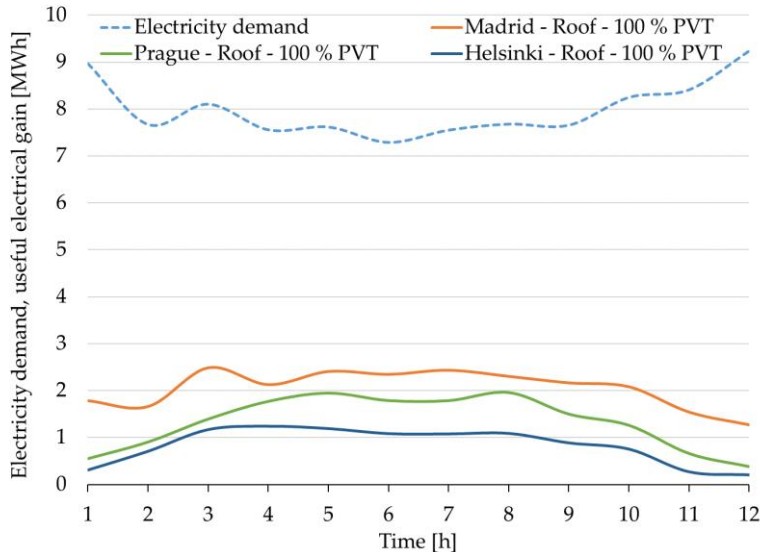

**Figure 16.** Monthly electricity production of PVT system and electricity demand (all three locations).

## 5. Conclusions

The simulation analysis of solar system in multifamily building for three different European locations has been performed. Mathematical model of a glazed PVT collector has been experimentally validated, implemented in TRNSYS, and used for the simulation study. New concept of glazed PVT collector with polysiloxane gel encapsulation of PV cells has been manufactured in separated and integrated alternative. The results of the simulation study have shown large potential to maximize utilization of incident solar radiation by PVT system compared to side-by-side installation of conventional technologies (PV and solar thermal collectors). The PVT system's electric yield ranged between 63 and 149 $kWh/m^2$. The increase in electricity production for the PVT system varied from 19% to 32%. The specific thermal yield of the PVT system ranged between 352 and 582 $kWh/m^2$. The increase in usable heat production differed between the façade and roof alternatives. The increase in thermal yield for the roof alternative ranged from 37% to 53%. The increase was much higher in the façade alternative and ranged from 71% to 81%. However, the specific thermal and electrical yield of façade PVT installation was lower in comparison with the roof PVT installation. The specific thermal yield of the PVT system for the façade alternative ranged between 295 and 481 $kWh/m^2$. The specific electrical yield of the PVT system for the façade alternative ranged between 63 and 96 $kWh/m^2$. In terms of worldwide pressure to reduce $CO_2$ emissions, PVT systems for multifamily buildings would play an important role in the decarbonisation process.

**Author Contributions:** Conceptualization, N.P. and T.M.; Data curation, N.P.; Formal analysis, T.M.; Methodology, T.M.; Software, N.P.; Validation, N.P.; Supervision, T.M.; Writing—original draft, N.P.; Writing—review and editing, N.P. All authors have read and agreed to the published version of the manuscript.

**Funding:** This research was funded by Ministry of Education, Youth and Sports within National Sustainability Programme I, project no. LO1605.

**Conflicts of Interest:** The authors declare no conflict of interest.

## Appendix A

**Table A1.** Parameters of PVT collectors used in simulation study.

| Parameters | Separate PVT | Integrated PVT | Unit |
|---|---|---|---|
| **Geometrical properties** | | | |
| Gross area | 1.65 | 1.56 | m$^2$ |
| Aperture area | 1.55 | 1.37 | m$^2$ |
| Gap between glazing and PVT absorber | 24 | 24 | mm |
| Gap between PVT absorber and frame | 5 | 5 | mm |
| Thickness of glass | 4 | 4 | mm |
| Thickness of insulation (back side) | 40 | 160 | mm |
| Thickness of insulation (lateral side) | 10 | 20 | mm |
| **Properties of the sheet and tube absorber** | | | |
| Thermal conductivity of the absorber plate | 350 | 350 | W/m.K |
| Thickness of absorber plate | 0.2 | 0.2 | mm |
| Length of riser pipes | 1.515 | 1.415 | m |
| Number of riser pipes | 20 | 18 | - |
| Distance between riser pipes | 50 | 50 | mm |
| Internal diameter of riser pipes | 7.2 | 7.2 | mm |
| **Thermal and optical properties** | | | |
| Normal solar transmittance of glass | 0.91 | 0.91 | - |
| Absorptance of PVT absorber | 0.90 | 0.90 | - |
| Front surface emissivity of PVT absorber | 0.30 | 0.85 | - |
| Back surface emissivity of PVT absorber | 0.85 | 0.85 | - |
| External surface emissivity of frame | 0.50 | 0.50 | - |
| Internal surface emissivity of frame | 0.50 | 0.50 | - |
| Thermal conductivity of glass | 0.80 | 0.80 | W/m.K |
| Outer surface emissivity of cover glass | 0.85 | 0.85 | - |
| Inner surface emissivity of cover glass | 0.85 | 0.85 | - |
| Thermal conductivity of insulation | 0.04 | 0.04 | W/m.K |
| Thermal conductivity of insulation (lateral side) | 0.04 | 0.04 | W/m.K |
| Thermal conductivity of lamination layer | 0.16 | 0.16 | W/m.K |
| Thickness of lamination layer | 2 | 2 | mm |
| Thickness of PV cells | 0.5 | 0.5 | mm |
| Thermal conductivity of PV cells | 149 | 149 | W/m.K |
| **Properties of PV part** | | | |
| Reference electrical efficiency of PV part without cover glass pane (related to PV area) | 16.3 | 16.3 | % |
| Temperature coefficient of PV cell efficiency | 0.44 | 0.44 | %/K |
| Gas layer between glazing | Argon | Argon | - |
| Packing factor | 0.62 | 0.60 | - |

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
