# Peer review of "Glazed Photovoltaic-thermal (PVT) Collectors for Domestic Hot Water Preparation in Multifamily Building"

_sustainability, doi:10.3390/su12156071_

Round 1

Reviewer 1 Report

Comments for the manuscript

Glazed Photovoltaic-Thermal Collectors for Domestic

 Hot Water Preparation in Multifamily Building”

The subject investigated in the paper is interesting and the methodology used and the results obtained may be proved useful in the photovoltaic/thermal applications. However, after a study of the manuscript, the following comments should be taken into account.

  1. The authors do not mention the hydraulic connection (series/parallel) for PVTs as well as for flat plate solar collectors. However, it plays an important role in their performance.
  2. Since the work focuses on DHW preparation, the authors do not mention the daily profile of thermal and electric load.
  3. Because the electrical characteristics of the PVT are different from those of the simple PV, the comparison of the results does not make sense. The question that arises is why the experimental determination of the characteristic curve for PVT was not made without the absorber and the glass cover? Thus, the electrical efficiency of the PV modules would be comparable.
  4. In addition, the comparison of thermal performance between the PVT and the PV/flat plate solar collector is questionable, as the authors are using a type of TRNSYS, which they report in line 155, that does not give “ reliable results”.
  5. Finally, the comparison of the results for the three cities is questionable, as the authors use the same slope. In solar systems, the slope of the solar collectors depends on the latitude of the location and can never be the same for all three cities.

In conclusion, and taking into account the aforementioned comments, the paper does not meet the requirements for publication in its present state.

Reviewer 2 Report

Some improvement is necessary in the following areas in order to be considered for publication:

==Review and edit for English grammar and paper structure (i.e. lines 45-60 , etc. --need better flow; long paragraphs, etc) throughout

-- Need to provide proper references (i.e. TRNSYS, UCEEB, etc). Discuss each reference individually, don't lump references

--Discuss some specific results in the abstract

--Maybe reference to a Figure early in the introduction is needed to better explain what is being studied

--maybe integrate section 2 into intro...some of that information should occur earlier

--spell out Table

--discuss other PVT systems

--better discussion of the design is needed with proper components, details and accuracy of the lights, meters, etc

-- please be consistent with number format  1,00 or 1.00

--provide details of the solar simulator used

--Discuss Figure 2 in more detail and annotate the figure. Setup is unclear

--Any effect of wind or other atmospheric conditions?

-- Don't use "We" in the text

--more descriptive titles necessary for figures and tables

--Table 3 parts list needs more description

--please specify building orientation  section 3.1, etc

--please specify heights for Figure 9 (floors, buildings, etc)

--Better Figure 10, Table 4, Table 5, etc and better explanation/discussion, better axes title

--Stronger and improved conclusions

Reviewer 3 Report

The article describes a prototype of a glazed PVT collector and its characterization for the installation integrated in the building envelope and the fixed installation on the roof. In the second part, using the TRNSYS simulation software, the electrical and thermal production (referring only to the DHW demand) is compared with the conventional systems (solar thermal and photovoltaic systems). The study is carried out considering three European locations.

Although the topic has been extensively covered in literature, the paper could still bring new knowledge and new food for thought, but in any case it must be strongly revised because it is lacking in some aspects, therefore it is considered essential to act on some points:

1) Intrduction should be expanded. There are other articles that analyze PVT systems by comparing them with conventional systems (https://doi.org/10.1016/j.jclepro.2019.02.038) or articles concerning the installation of PVT systems on the facade (https://doi.org/10.1016/j.renene.2017.11.060), so it would be appropriate to quote them and motivate what adds this work to previous knowledge.

2) The method used must be explained in greater detail, it is not clear how the thermal energy produced is calculated. In addition, the thermal and electrical energy produced should be related to the building's needs, which has not been described.

3) In the comparison of the energy produced by the PVT and that produced by conventional plants, the use of 50% of the available surface with photovoltaic panels and 50% with solar panels is assumed. This choice must be motivated, because as described by (https://doi.org/10.1016/j.jclepro.2019.02.038) the maximum energy produced by separate conventional plants is not obtained considering the 50:50 ratio.

4) The results should be extended. Regarding the thermal analysis it would be appropriate to add the analysis of the thermal level reached and some graphs on the fraction of demand satisfied as the period of the year changes, because there can be periods where excess energy is produced and others where the satisfaction factor of the demand is very low. In the same way, the coverage factor of electricity demand should be clarified as the period of the year changes.

5) An economic analysis for the various locations could finally complete the comparison work of the PVT systems with the conventional ones and the study considering various locations.

6) Based on the previous comments, the conclusions of the paper should be reviewed.

Round 2

Reviewer 1 Report

I agree with the revised manuscript.

Author Response

Thank you for the revision.

Reviewer 2 Report

While this is an improvement over the last version some additional improvement needs to be done to the manuscript. English grammar edit and review is needed.

Additionally several points from the previous review were not completed

==Review and edit for English grammar and paper structure (i.e. lines 45-60 , etc. --need
better flow; long paragraphs, etc) throughout
I tried to improve.

Still needs improvement

-- Need to provide proper references (i.e. TRNSYS, UCEEB, etc). Discuss each reference   STILL NEEDS TO BE DONE
individually, don't lump references
I edited some references but there are still some lumped references because these papers have
very similar topic.

Discuss some important point from each reference  STILL NEEDS TO BE DONE

--Discuss some specific results in the abstract
I added more results.  OK

--Maybe reference to a Figure early in the introduction is needed to better explain what is
being studied
I am not sure which figure do you mean.  SOMETHING TO HELP YOU BETTER EXPLAIN THE NOVELTY OF THE STUDY
--maybe integrate section 2 into intro...some of that information should occur earlier
It is already there some info about the concept of PVT collector with encapsulation of PV
cells into gel. I added more info about the SDHW system and review about this topic.

MOVE SECTION 2 INTO THE INTRO .  SOME OF THE INFO IN THAT SECTION NEEDS TO BE INTRODUCED SOONER THAN SECTION 2

--spell out Table
I am not sure which table do you mean.

WHEN YOU REFERENCE TABLES  SPELL OUT TABLE DON'T  USE TAB

--discuss other PVT systems
I added into introduction if it is enough.OK

--Any effect of wind or other atmospheric conditions?
Boundary conditions of the test are deeply described in the ISO 9806. I added in the text more
info related to the test.

PROPER REF TO ISO 9806 NEEDS TO BE MADE AND YOU SHOULD DISCUSS THE EFFECTS OF CLOUDS, WIND, ETC

Reviewer 3 Report

The review process allowed a general improvement of the article, with greater clarity in the introductory part, in the method used and in the conclusions. The system operating scheme (figures 9) could be further improved by inserting the operating conditions.
